# Transcriptional Spatial Profiling of Cancer Tissues in the Era of Immunotherapy: The Potential and Promise

**DOI:** 10.3390/cancers12092572

**Published:** 2020-09-09

**Authors:** Sanjna Nilesh Nerurkar, Denise Goh, Chun Chau Lawrence Cheung, Pei Qi Yvonne Nga, Jeffrey Chun Tatt Lim, Joe Poh Sheng Yeong

**Affiliations:** 1Yong Loo Lin School of Medicine, National University of Singapore, Singapore 117597, Singapore; e0199368@u.nus.edu; 2Institute of Molecular and Cell Biology (IMCB), Agency of Science, Technology and Research (A*STAR), Singapore 169856, Singapore; denise_goh@imcb.a-star.edu.sg (D.G.); yvonne_nga@imcb.a-star.edu.sg (P.Q.Y.N.); limct1@imcb.a-star.edu.sg (J.C.T.L.); 3Duke-NUS Medical School, Singapore 169857, Singapore; e0316125@u.duke.nus.edu; 4Department of Anatomical Pathology, Singapore General Hospital, Singapore 169856, Singapore; 5Singapore Immunology Network (SIgN), Agency of Science, Technology and Research (A*STAR), Singapore 138648, Singapore

**Keywords:** heterogeneity, clonal diversity, transcriptomics, in situ hybridization, digital spatial profiling, cancer, biomarkers, immunotherapy

## Abstract

**Simple Summary:**

In recent years, immunotherapy has emerged as a pillar in the fight against cancer. However, the heterogeneity within the tumor microenvironment poses challenges to the efficacy of immunotherapy treatment strategies and may contribute to treatment resistance, progression and relapse. Hence, researchers have used transcriptional spatial profiling techniques to uncover the complex cancer immune profile while retaining critical spatial information of different cell types. This would help identify the source of tumor heterogeneity and uncover pathogenic mechanisms, potential drug targets and novel biomarkers. In this review, we discuss various approaches for transcriptional spatial profiling of cancer tissues such as in situ hybridization, digital spatial profiling and an emerging technology known as Visium Spatial Gene Expression Solution. We highlight the strengths and limitations of the different technologies and the potential and promise they hold in the advancement of cancer immunotherapy.

**Abstract:**

Intratumoral heterogeneity poses a major challenge to making an accurate diagnosis and establishing personalized treatment strategies for cancer patients. Moreover, this heterogeneity might underlie treatment resistance, disease progression, and cancer relapse. For example, while immunotherapies can confer a high success rate, selective pressures coupled with dynamic evolution within a tumour can drive the emergence of drug-resistant clones that allow tumours to persist in certain patients. To improve immunotherapy efficacy, researchers have used transcriptional spatial profiling techniques to identify and subsequently block the source of tumour heterogeneity. In this review, we describe and assess the different technologies available for such profiling within a cancer tissue. We first outline two well-known approaches, in situ hybridization and digital spatial profiling. Then, we highlight the features of an emerging technology known as Visium Spatial Gene Expression Solution. Visium generates quantitative gene expression data and maps them to the tissue architecture. By retaining spatial information, we are well positioned to identify novel biomarkers and perform computational analyses that might inform on novel combinatorial immunotherapies.

## 1. Introduction

Immunotherapy has emerged as a promising and powerful pillar in the fight against cancer: These therapeutics activate the patient’s immune system to kill cancer cells. Current immunotherapeutic strategies are based on cancer vaccines, cytokines (such as interleukin-2), adoptive cell transfer (ACT), and immune checkpoint inhibition [1,2]. These treatments have shown great promise, and immune checkpoint inhibitors targeting the programmed cell death-1/programmed cell death-ligand 1 (PD-1/PD-L1) pathway in particular, have now been approved as first- or second-line treatments for melanoma, lymphoma, and other malignancies [3]. However, only a minority of patients positively respond to treatment [4]; some patients ultimately develop resistance [5] and/or even suffer adverse reactions [6] and autoimmune toxicity [7] as a result of the treatment. The reason for this poor result in a subset of patients is partly attributed to the composition of the dynamic tumour microenvironment (TME) [8,9,10]. Here, a complex interplay between tumour cells, infiltrating immune cells, and the stroma as well as the location and density of immune cell populations affects disease progression and responses to therapy [11,12]. This paradigm has spurred the development of technologies that can characterize the cancer immune profile while retaining spatial information of the various cell types. Spatial information is critical, as visualizing the interplay between the tumour and various cell groups that mediate immune surveillance will improve our understanding of pathogenic mechanisms and potential drug targets. Spatial context, including cell–cell distances and feature heterogeneity, can also be correlated with clinical outcomes to identify predictive biomarkers for responses to immunotherapy [13]. Meanwhile, analysing how current immunotherapeutic strategies alter the TME architecture and immune context can help guide future treatment approaches [14]. Together, we are better able to identify novel biomarkers, potential drug targets and pathogenic mechanisms. 

Multiplex immunohistochemistry/immunofluorescence (mIHC/IF) is one commonly used tool that enables the simultaneous detection of up to 40 markers of interest in a single tissue sample [15,16]. This approach was better able to predict the cellular response to PD-1/PD-L1 therapy compared to analyses of the tumour mutational burden (TMB) or gene expression profile (GEP) alone [17]. This is because mIHC/IF has a spatial component that provides information on the TME architecture and co-expression of multiple cellular markers, unlike TMB analyses, which only provide quantitative measurements of mutations in tumour cells [18], and GEP, which only measures mRNA transcript levels of immune-related genes or signatures. [19] However, despite these promising findings, multiplexed spatial analyses are limited by the number of markers that can be detected at any one time, compared to genomic techniques, which can provide a holistic view of the entire genome or transcriptome [20]. As a result, we tend to bias our selection of markers and thus have limited capacity for spatial analyses that follow a hypothesis-driven rather than an unbiased data-driven approach. Next-generation sequencing (NGS) has circumvented this limitation of mIHC/IF and enabled higher throughput whole-genome or whole-transcriptome sequencing compared to conventional mIHC/IF techniques [21,22]. However, such sequencing performed on bulk tissues or single cells after enzymatic dissociation comes at the sacrifice of critical spatial information [23,24]. 

Given these limitations, researchers and developers have focused efforts on finding a one-stop solution that offers both the breadth and depth of sequencing coverage of biomarkers as well as adequate resolution and spatial information. Spatial transcriptomics (ST) holds great promise in this area, providing researchers with the ability to identify novel biomarkers and insight into the dynamic interplay between tumour cells, adipose tissue, vessels, tertiary lymphoid structures, and the stroma in the TME [25]. In this review, we outline the features of the various technologies that are available for such transcriptional spatial profiling in cancer tissues and the potential and promise they hold in the advancement of cancer immunotherapy. 

## 2. In Situ Hybridization

In situ hybridization (ISH) is a molecular technique to visualize specific DNA or RNA molecules in cells or tissues. ISH is based on the complementary nature of DNA/DNA or DNA/RNA double strands and the hybridization of a labelled nucleic acid probe onto the target in situ. In this way, we can obtain useful spatial information. Traditionally, nucleic acid probes are attached to a radioactive label [26,27,28]; however, this approach has been largely replaced by a fluorochrome. The result is fluorescence in situ hybridization or FISH. FISH itself has been further developed into techniques known as multiplexed single-molecule FISH (smFISH), which is able to simultaneously detect approximately 10,000 genes and around 70,000–100,000 RNA molecules per cell at a single-molecule resolution. The basics of each technique are outlined below. 

### 2.1. Fluorescence In Situ Hybridization 

FISH is a useful clinical tool for detecting microorganisms, diagnosing solid and haematological cancers, and guiding cancer treatments. For example, FISH has been conventionally used to detect the BCR-ABL1 t(9;22) translocation in chronic myeloid leukaemia and many fusion genes in various cancers [29,30]. FISH has also been used to confirm human epidermal growth factor receptor 2 (HER2) gene amplification in breast cancer and thus to identify patients that are most likely to benefit from trastuzumab, a monoclonal antibody treatment against HER2 [31]. Another important example is the detection of the echinoderm microtubule-associated protein-like 4 and anaplastic lymphoma kinase (EML4-ALK) fusion gene in non-small cell lung cancer; this is now a routine clinical test, the results of which indicate the use of a targeted therapy known as crizotinib [32,33].

As more immunotherapies are developed and approved, researchers have sought to use FISH to predict responsiveness to immunotherapy in cancer. Not surprisingly, PD-L1 status is commonly investigated but its predictive and prognostic value are variable [34,35,36]. In addition, FISH might be useful for predicting responsiveness to intravesical bacillus Calmette–Guerin immunotherapy in bladder cancer [37,38]. To extend the effectiveness of FISH, we can combine FISH with IHC [39,40] or IF [41,42] to detect RNA and protein simultaneously in different cell types to better characterize TME. Despite these promising results and uses of FISH, conventional FISH largely detects cytogenetic aberrations at the DNA level; it fails to address the complexity of the TME and intra-tumour heterogeneity arising from differential mRNA and protein expression. High-throughput methods are still needed to study individual tumours on a larger scale.

### 2.2. Single-Molecule FISH and RNAscope 

In an attempt to address the limitations of conventional FISH, research efforts have shifted from studies of DNA to studies of single-molecule RNA, using high-throughput methods. The resulting technique, known as single-molecule FISH (smFISH), allows researchers to visualize and quantify individual mRNA molecules and characterize the spatial pattern of endogenous gene expression [43,44,45,46]. By targeting cellular mRNA instead of DNA molecules, smFISH has become a robust tool to assess intra-tumour transcriptional heterogeneity [47,48,49].

RNAscope (Advanced Cell Diagnostics, Hayward, CA, USA) is a commercialized ISH-based technology that uses branched DNA and signal amplification to achieve better sensitivity and specificity than conventional FISH: By this approach, we can detect and quantify low-abundance mRNA [50] while suppressing background noise [51]. This method permits the detection of up to 12 different RNA targets and can be conveniently combined with IHC and/or IF to simultaneously study RNAs and proteins in an automated manner [52,53,54]. A major advantage of RNAscope over other FISH-based technologies is that >13,000 RNA probes have already been designed and validated with commercially established protocols; it is thus a time-saving and user-friendly approach for use in research and clinical laboratories. 

RNAscope has already been widely used in various disciplines, including in the context of infectious diseases, cancers, immunotherapy, inflammation, and neurosciences [55,56,57,58]. In particular, it is a robust alternative to IHC to evaluate the expression of immune checkpoints, such as PD-L1, in various solid cancers [59,60,61,62,63,64]. By detecting a specific RNA of interest, RNAscope has shed light on the TME, the mechanisms of immune escape [65,66,67,68,69,70,71], and new predictive and prognostic cancer biomarkers [72,73].

In the context of immunotherapy, RNAscope has played valuable roles in understanding chimeric antigen receptor (CAR)-T cell therapy, a technique by which T cells are genetically engineered to produce receptor proteins that target cancer cells. RNAscope has been used to assess the specificity of target gene expression and to track the distribution of CAR-T cells in a xenograft mouse model [57]. Extending to human samples, Bu et al. validated the expression of B-cell maturation antigen (BCMA) as a target for CAR-T cell immunotherapy in multiple myeloma [74] while O’ Rourke et al. tracked infused CAR-modified T cells against glioblastoma using RNAscope [75]. Overall, despite its limited multiplexed capacity and low target throughput, RNAscope is still a viable option for researchers and clinical laboratories to study the expression of targeted RNA species in cancer: The sensitivity and convenience is so high that studying only a handful of RNA targets is useful.

### 2.3. Multiplexed smFISH

Although we can gain increased sensitivity and specificity from technologies, such as RNAscope, we ultimately need FISH-based technologies that permit a high-throughput transcriptomic analysis to better characterize rare populations of cells and cell types that display a distinctive gene expression profile. New technologies, such as multiplexed error-robust FISH (MERFISH) (Vizgen, Cambridge, UK) and sequential FISH (seqFISH), not only provide improved RNA quantification, signal amplification, and detection but also, more importantly, offer the opportunity of image-based transcriptome analysis by massively increasing the number of RNA species that can be detected at a given time [76,77].

Modified from smFISH, MERFISH employs a barcode-based combinatorial labelling approach followed by multiple rounds of hybridization to ensure a high level of brightness of the fluorescence signal and a high number of RNA species that can be detected at one time (Figure 1) [78]. Impressively, MERFISH can achieve near genome-wide profiling and a detection efficiency of >90%, as demonstrated using a modified error-robust protocol in human osteosarcoma cells [79,80]. Compared to conventional FISH, MERFISH also provides the additional benefit of being able to quantify individual RNA molecules at a low abundance. 

seqFISH is another multiplexed smFISH technique that is based on sequential rounds of barcoded hybridization labelling [81]. Interestingly, it provides a spatial resolution at the sub-diffraction limit that is superior to other RNA-profiling techniques [77]. For example, seqFISH was used to image >10,000 mRNA species in mouse embryonic stem cells and brain tissues with high accuracy and resolution [77,82]. Zhou et al. demonstrated that seqFISH is a powerful tool to study and gain a snapshot of the regulatory gene expression dynamics in T-cell maturation [83]. More recently, Voith von Voithenberg et al. combined microfluidics technology with multiplexed smFISH to study tumour heterogeneity in breast cancer; they demonstrated that multiplexed smFISH can be further optimized from different angles [84].

smFISH and multiplexed smFISH are appealing alternatives to single-cell RNA-sequencing to study and quantify cellular RNA as it provides subcellular spatial information at the single-cell and even single-molecule level. Yet, despite its promise, multiplexed smFISH-based technologies are not yet widely used in either translational research or in clinical settings due to the complex probe design, validation, image analysis, and decoding [85]. It is generally more convenient to use non-multiplexed FISH, quantitative PCR, IHC, and IF to study single gene expression at the mRNA or protein level, especially when the number of genes being studied is small, such as a set of prognostic markers. Another limitation is that, because of the nature of sequential hybridization, the total imaging time adds up to a minimum of 18 h excluding the initial probe hybridization time of 36–48 h [76], resulting in an overall low throughput compared to other library-based technologies, like digital spatial profiling and Visium (Table 1), which we will discuss later. In addition, multiplex smFISH techniques only assess one type of analyte, such as RNA, in predominantly fresh-frozen tissues [86]. Emerging technologies, such as digital spatial profiling, can assess both protein and RNA levels in fresh-frozen tissues and standard formalin-fixed paraffin-embedded (FFPE) tissues that are routinely used by pathologists [87,88].

## 3. Digital Spatial Profiling 

Digital spatial profiling (DSP) is a high-plex spatial profiling method that overcomes the key limitations of multiplexed smFISH techniques, including prolonged experimental times that decrease sample throughput, the limited capacity to only assess a single analyte type (RNA or protein), a lack of FFPE compatibility, and the lack of a commercial integrated system. DSP uses oligonucleotide detection technologies to quantify protein or RNA levels in FFPE tissue samples (Figure 2). Oligonucleotide tags are conjugated using a photocleavable UV light-sensitive linker to primary antibodies or RNA probes, for protein or RNA profiling, respectively. Firstly, the FFPE tissue section undergoes antigen retrieval and incubation with oligo-labelled primary antibodies or proteinase K digestion and incubation with an RNA probe cocktail. The slides are then stained with fluorescently labelled antibodies (up to four markers) to visualize features of interest and enable region of interest (ROI) selection on the tissue sample. The user-defined ROI is illuminated with UV light, releasing the photocleavable oligonucleotides from the antibodies or RNA probes. These oligonucleotides are collected in a microcapillary tube and transferred to a microtiter plate for quantitation. This process is repeated for the next ROI. Once all ROIs have been processed, spatially resolved pools of oligonucleotides are hybridized to capture and reporter probes with unique fluorescent barcodes that can be digitally counted using the nCounter analysis system. Alternatively, next-generation sequencing (NGS) may be used as a readout, where reads are processed into digital counts that are mapped back to the ROI to allow spatial profiling of the ROI [20,87].

Unlike sequential hybridization techniques, such as MERFISH, which have prolonged experimental times of up to 48 h for one slide (including probe hybridization) [76], DSP offers a more efficient workflow that generates results from 10–20 tissue sections or up to 384 regions of interest within 48 h (Table 1) [87]. In addition, DSP can detect up to 96 proteins or 1400 mRNA simultaneously [20,87] as compared to multiplex smFISH, which only profiles RNA. This feature is particularly relevant for cancer immunotherapy, as discrepancies in mRNA and protein expression patterns can be used to elucidate post-transcriptional regulation and post-translational modifications contributing to protein instability and affecting prognosis and response to therapy [20,89]. DSP is also particularly attractive for clinical use as it offers a commercial integrated system that is highly automated, and optimized workflows, validated assays, and data analysis software are available. In addition, it also preserves the integrity of tissue samples, allowing precious samples to be stored and used for further analysis in the future. 

DSP has been used widely in the field of immunotherapy. Jeyasekharan et al. used DSP to assess the immune microenvironment of patients with diffuse large B-cell lymphoma (DLBCL) who had been treated with chemo-immunotherapy. Using a customized immune panel of 36 antibodies, the researchers found that tumour infiltration by M2 macrophages (CD163 and CD68) had a significant negative impact on prognosis. This finding provided the basis for further treatments targeting tumour-infiltrating macrophages [90]. DSP has also been explored in the area of immune checkpoint blockade therapies, including anti-PD-L1 and anti-PD-1 therapy [91]. Specifically, researchers found that DSP could be used to quantify PD-L1 expression objectively and accurately in a standardized cell line index tissue microarray (TMA). The concordance of PD-L1 measurements with other routinely used techniques, such as quantitative immunofluorescence (QIF), was high, with coefficients > 0.9 while providing the additional benefit of high reproducibility that was independent of the slide storage time [89]. Existing commercial PD-L1 IHC assays are semi-quantitative and require scoring by trained pathologists, which may introduce error due to inter-observer variability [92,93]. DSP might thus be used as a companion diagnostic tool that provides standardized, quantitative, and objective assessments of PD-L1 protein expression within spatially defined compartments in the tumour microenvironment.

In a separate study, DSP was shown to successfully identify >20 biomarkers that predict responses to immunotherapy in patients with melanoma. The most notable finding was that PD-L1 expression in CD68-positive cells (macrophages) rather than in tumour cells was integral to determining progression-free survival, overall survival, and treatment responses in melanoma patients [94]. Others have used DSP to explore responses of non-small-cell lung cancer (NSCLC) treatment with checkpoint inhibitors. By using tumour samples from NSCLC patients and studying immune infiltration in four different compartments (tumour, macrophages, leukocytes, and non-immune stroma), researchers found that DSP could identify prognostic biomarkers predicting the response to checkpoint therapy. For example, high CD56+ cell counts in the stroma were associated with improved survival while high CD127 levels in the tumour compartment were associated with immunotherapy resistance [95]. Thus, it seems that DSP has the potential to be an accurate and reproducible tool for determining patient prognosis following checkpoint inhibition. The usefulness of DSP is not just limited to fresh-frozen or FFPE tissue samples; indeed, one study showed how DSP could be extended to bone marrow trephine samples with unprecedented high-plex spatial profiling of the bone marrow microenvironment [96]. This finding means that DSP might serve to identify biomarkers and potential drug targets specific to haematological malignancies. 

Although DSP shows promise in the context of immunotherapy research, some key limitations remain. Firstly, DSP requires the selection of regions of interest (ROI) for analysis. This is mostly an automated process that allows great flexibility in the types of ROI selection, such as geometric, gridded, rare cell population profiling, and segmentation into the tumour and TME [20,89]. While this is particularly useful in interrogating the TME and features of interest, it prevents whole-tissue analysis especially in larger excisional biopsy samples and may lead to a biased hypothesis-driven sample analysis [88,97]. In addition, DSP has poor single-cell resolution, requiring at least 10 cells in an ROI to generate adequate counts [96,98]. This restriction might limit the effectiveness of DSP when analysing tumour regions with low cellularity. Secondly, DSP only provides images based on its ‘morphology kit’ for no more than four colours. Multiplexing only provides numerical data in the form of counts that are detected within the ROI. As there is no reconstructed tissue image, it is not possible to ascertain staining quality and there is a loss of critical spatial information [97]. Furthermore, the multiplexing capacity is limited to up to 1400 genes [87], which is comparatively less than other tools, such as MERFISH and Visium (discussed below), which have multiplexing capacities of 10,000 and 100,000 genes, respectively (Table 1). There have been efforts to overcome this limited multiplexing capacity by integrating DSP with NGS readout, which theoretically might lead to an unlimited multiplexing capacity, but studies have yet to demonstrate this [88,99]. This lack of sequencing information offered by DSP has thus spurred the development of spatial transcriptomics (ST), which enables unprecedented full-transcriptome profiling while retaining spatial context. 

## 4. Spatial Transcriptomics 

During single-cell RNA sequencing, spatial information is lost as tissues are often homogenized to obtain an averaged overview of the transcriptome [100,101]. Although this technique is widely used to explore gene expression profiles at a single-cell level, it confers a low capture efficiency and sequencing coverage, as well as a high rate of dropout events, which together can impede downstream data analysis and interpretation [101,102,103]. Recently, the emerging field of spatial genomics entered the arena, with the development of a technique pioneered by a company aptly named “Spatial Transcriptomics” (Stockholm, Sweden). This technique enables quantitative visualisation and analysis of the transcriptome within intact tissue sections with the use of spatially barcoded oligo-deoxythymidine microarrays [100,101]. Here, unique positional barcodes are introduced onto glass slides to preserve spatial positioning within the tissue architecture before proceeding with the RNA sequencing process [104] (Figure 3). 

This novel technique was first demonstrated on the mouse olfactory bulb, and it follows a standard workflow as follows: Tissue sectioning, fixation, haematoxylin and eosin (H&E) staining, bright-field imaging, tissue permeabilization, cDNA synthesis, tissue removal, probe release, library preparation, sequencing, data processing, data visualization, and analysis [101,105,106,107]. Notably, a distinctive feature of this workflow is the ability to generate an on-slide cDNA library with preserved spatial information, making it possible to visually map the gene expression profile to its corresponding tissue morphology [105]. This possibility encourages the identification of novel gene targets and the early detection of premalignant tissue areas that might not be identifiable by pathologists.

Various success stories applying spatial transcriptomics (ST) techniques have been reported in the literature, based on identifying unique gene expression profiles in tissue biopsies prior to histopathological annotations. Data analysis of breast cancer [101], prostate cancer [108], and cutaneous malignant melanoma [109] biopsies by ST have revealed an unprecedented level of intra- and inter-tumoral heterogeneity, as well as distinct differences in gene expression profiles between the annotated tumour area and periphery that were not evident through RNA sequencing analysis and/or standard morphological annotations. Moreover, in vivo experimentation utilising this technique has identified the induction of IL-6 signalling by repopulating microglia, which might have value in the therapeutic context [110]. Of course, histopathological annotations and single-cell RNA sequencing can identify aberrant tissue morphologies and confirm the presence of genetically distinct cell populations, respectively; however, ST can highlight distinct spatial regions based on gene expression profiles.

To harness the potential of ST, researchers recently developed an analytical approach known as multimodal intersection analysis (MIA). MIA incorporates datasets generated from single-cell RNA sequencing and ST techniques to produce an unbiased map of transcripts across the tissue footprint at a cellular level. As a proof-of-concept, MIA was performed on a pancreatic ductal adenocarcinoma dataset and revealed specific cell type and subpopulation enrichment across spatially restricted regions that was previously unknown or undetectable [100].

### Visium Spatial Gene Expression Solution

Based on the concept pioneered by Spatial Transcriptomics, Visium Spatial Gene Expression Solution (10× Genomics, USA) was recently released with enhancements, such as a higher resolution and increased sensitivity, compared to the first iterations of the ST technique [111]. In the context of human squamous cell carcinoma [112], single-cell RNA sequencing revealed a distinct subpopulation of tumour-specific keratinocytes (TSKs) with signature genes associated with epithelial-mesenchymal transition and invasive behaviours. The first-generation ST technique identified clear TSK clusters at the tumour leading edges, including TSK marker *MMP10*, in each patient. Using the enhanced Visium technique, additional spot transcriptomes were identified and as such, enrichment of endothelial and cancer-associated fibroblast-associated transcripts at the stroma were demonstrated. This finding reveals and supports a fibrovascular niche surrounding TSKs in the tumour microenvironment. 

Visium is a well-built platform for in-depth investigations of diseases that are associated with tissue structure and function. This is because it can identify tissue regions with aberrant gene expression, allowing for the discovery of novel biomarkers within an area. For example, the laminar organization of the human cerebral cortex is highly complex; studies of neurological disorders have proposed that differences in pathology and gene expression profiles are localized to specific cortical layers [113]. By applying Visium to the human dorsolateral prefrontal cortex, researchers have successfully defined the spatial topography of gene expression profiles within the tissue, identified several formerly underappreciated layer-enriched expression profiles, and verified laminar enrichment of several genes in specific cortical layers [114]. These findings suggest that as well as cancer immunotherapy, Visium could also be applied to neurological disorders.

The Visium software programs for data analysis and visualisation of the generated cDNA library to achieve multidimensional datasets are provided by 10× Genomics (Space Ranger and Loupe Browser). With an end-to-end workflow, Visium can be easily integrated into existing lab infrastructure; no specialised equipment is required other than a cryostat, microscope, and sequencer [105]. The 10× Genomics platform has been used to profile tumour-associated macrophages (TAMs) from patient biopsies, where it helped to quantify the main subpopulations. By mapping glioma structures, it was revealed that microglia take the lead in tumour infiltration while blood-derived TAMs are enriched near blood vessels. As a result, a negative correlation between blood-derived TAMs and low-grade gliomas was reported. This finding supports the notion that macrophage ontogeny is critical to shaping macrophage activation in the glioma microenvironment [115].

Particularly in the field of spatial genomics, 10× Genomics’ Visium is a promising platform with the capacity to construct high-resolution microscopic images with gene expression data aligned to the tissue footprint. It allows flexibility to analyse spatial gene expression from different angles within a single experiment and because it embraces the ST technique, users can perform multiple data generation without losing valuable information as original tissue transcripts are conserved on the slide [114]. Apart from the software programs provided by 10× Genomics, Itai Yanai and his team proposed that Visium might also be compatible with MIA [116]. 

Despite its great promise, there are limitations to Visium that must be noted. Although the recommended optimal tissue thickness is 10 µm, this value is dependent on the tissue type and composition. 10× Genomics have produced a support site that provides users with an updated list of compatible tissues and the corresponding thicknesses; to date, there are currently 4 rat tissues, 20 mouse tissues, and 19 human tissues listed, as well as 4 tissues planned for further optimisation [117]. Users are also recommended to run a one-time optimisation experiment for every new tissue type, as tissue permeabilization conditions vary between tissues, species, and even laboratories. We anticipate that 10× Genomics’ Visium Spatial Tissue Optimization Slide and Reagent Kit will aid in ensuring tissue compatibility and a better workflow. Furthermore, although Visium has only been validated in fresh-frozen specimens [105], early studies suggest that it can be used for genome-wide spatial profiling in FFPE specimens as well [118]. Lastly, as part of ST enhancement, each tissue capture area contains ~5000 spots with an individual spot size of 55 µm; this generates a cell resolution of 1–10 cells per spot depending on the tissue type and thickness [119]. While some might find this resolution satisfactory, extra precaution should be taken when analysing spots at a boundary. One suggestion is to integrate MIA into the workflow to allow read-outs at a cellular level. Regardless of these limitations, because Visium can promptly identify aberrant gene expression profiles and detect emerging hallmarks of cancer initiation and progression without losing spatial information, we feel that the potential of Visium prevails over the aforementioned technologies. 

## 5. Conclusions and Future Perspectives 

Transcriptional spatial profiling techniques have rapidly evolved over recent years. From early tools, such as FISH-based technologies, that allow for the analysis of a few gene targets to recent developments, such as Visium, that allow unprecedented whole-transcriptome analysis, we are now able to characterize the cancer immune profile with high-throughput technologies while retaining critical spatial information and resolution. With such rapid advancements in spatial profiling techniques, the relevance of traditional multiplex IHC/IF may come into question. Multiplex IHC/IF has been used extensively in both the research and clinical settings to simultaneously detect multiple target proteins in the same tissue sample. Although there are numerous commercialized multiplexed tissue imaging techniques available, such as multiplexed ion beam imaging (MIBI) [120] and imaging mass cytometry (IMC) that evaluate up to 40 biomarkers at any one time [121], these techniques still lack the high multiplexing capacity and spatial information offered by other transcriptional spatial profiling techniques. However, multiplex IHC/IF still retains key advantages in cancer tissue analysis that are worth remembering. Firstly, proteins are functional molecules and hence, gene expression that is evaluated using transcriptional spatial profiling techniques might not necessarily correlate with protein expression. This is particularly relevant in PD-L1 checkpoint inhibitor therapy, where PD-L1 IHC is the primary biomarker assay currently used for selection of patients for checkpoint inhibitor therapy as increased PD-L1 tissue expression is associated with improved survival rates [122,123]. Interestingly, in two separate studies [124,125] that used transcriptomic technology to predict the response to anti-PD1 therapy, it was found that genes involved in the checkpoint pathway, such as PD-L1 and CD8A/B, showed no significant association with response to therapy [19]. Instead, inflammatory tumour phenotypes [124] and the expression levels of metabolic-related genes [125] were found to predict an anti-PD1 response. Thus, while transcriptional spatial profiling techniques may enable extensive sequencing up to the entire genome level, one must consider that not every mRNA transcript necessarily leads to translation and synthesis of biologically active proteins that contribute to tumour initiation, progression, and therapeutic response. This may be attributed to the heterogeneity in signalling pathways, post-translational modifications, and protein isoforms, which limits the utility of mRNA abundance as a proxy for protein abundance and activity [126,127,128,129]. Therefore, protein-based technologies, such as multiplex IHC/IF, remain integral for downstream analysis after transcriptomics studies as they serve as an excellent endpoint to validate protein function after identifying genetic targets. Multiplex IHC/IF is not redundant, and instead, should be viewed as a complementary tool to the transcriptional spatial profiling techniques discussed above. 

Despite being relatively new, transcriptional spatial profiling technologies have been explored widely in cancer immunotherapy. FISH and RNAscope are useful clinical tools for the diagnosis and prognostication of solid and haematological cancers [130,131,132]. Newer techniques, such as MERFISH and Visium, have overcome some of the limitations to their multiplexing capacity by enabling bulk transcriptome analysis with an unprecedented level of resolution and sensitivity [79,133]. The increasing accessibility of such techniques is exciting for the field of cancer immunology, as they enable the discovery of novel biomarkers that serve to predict responses to immunotherapy and permit personalized treatment approaches based on the heterogeneity of their unique TME [108,109]. These spatial profiling techniques can potentially be combined with dimensionality-reduction techniques, such as uniform manifold approximation and projection (UMAP), to visualize the immune landscape of the tumour microenvironment (Figure 4a). This will provide critical information about the immune cells surrounding the tumour border and stroma (Figure 4b) and can be correlated with clinical outcomes to determine predictive biomarkers. For instance, as intra-tumoral tissue hypoxia can also contribute to the heterogeneity of the tumour microenvironment, visualization of the hypoxia gradient (Figure 4c) is critical for determining treatment resistance and prognosis. Thus, overlaying the UMAP and hypoxia gradient (Figure 4d) can help provide a more holistic visualization of the heterogenous tumour microenvironment. 

Going forward, DSP offers spatial profiling and digital characterization of mRNA expression but remains limited by the number of gene targets that can be investigated simultaneously. However, its operation via a commercial platform with an optimized pipeline for both protein and RNA targets is likely to be attractive to some users [134]. Although Visium is relatively new to the market, 10× Genomics has recently released a new protocol that allows for IF staining instead of the conventional H&E staining. By combining whole-transcriptome analysis with protein detection in this way, users can spatially map cell populations and their gene expression profiles within tissues by visualizing co-localized protein and gene expression simultaneously [135]. Through the constant improvements made within a short time, Visium holds great potential to provide new insights into disease pathology and clinical translational research. 

Given the variety of evolving transcriptional spatial profiling techniques available to researchers, it is important that one considers both the technical characteristics of the technology, including the spatial resolution, sensitivity, specificity, and tissue type, as well as practical considerations, such as the cost, compatibility with available resources, and turnaround time. Ultimately, researchers must carefully consider their research questions and select an appropriate technology that closely aligns with their research and clinical goals. 

## Figures and Tables

**Figure 1 cancers-12-02572-f001:**
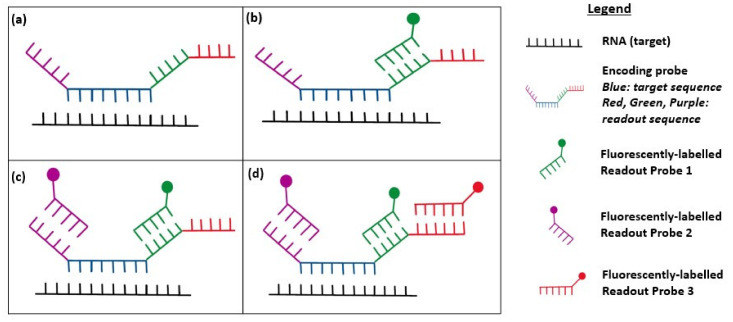
Diagram showing multiplexed error-robust FISH (MERFISH) imaging. (**a**) Diagram depicting the encoding probe hybridized to the cellular RNA; (**b**) After the first round of hybridization, readout probe 1 (green) binds to the complementary readout sequence on the encoder probe and is fluorescently labelled (green circle); (**c**) After round 2 of hybridization, readout probe 2 (purple) binds to the encoding probe; (**d**) This process is repeated with N rounds of hybridization, whereby readout probe N (red) will bind to the readout sequence on the encoding probe.

**Figure 2 cancers-12-02572-f002:**
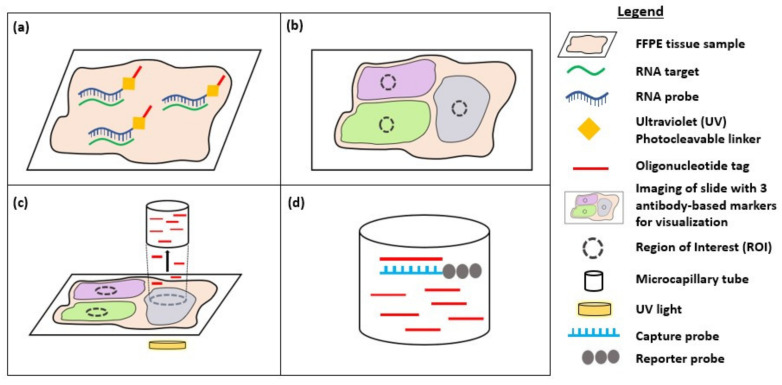
Diagram showing digital spatial profiling (DSP) for detection of RNA targets. (**a**) After the FFPE tissue sample undergoes proteinase K digestion, it is incubated with RNA probes that bind to target RNA in the sample. Each RNA probe is conjugated to an oligonucleotide tag via an ultraviolet (UV) photocleavable linker; (**b**) The slide is stained with 3 antibody-based markers (up to a maximum of 4 markers) and imaged to visualize tissue morphology and enable selection of regions of interest (ROI); (**c**) Illumination of the ROI with UV light causes the release of photocleavable oligonucleotides from the RNA probes. The oligonucleotides are collected in a microcapillary tube and transferred to a microtiter plate; (**d**) Spatially resolved pools of oligonucleotides bind to a target-specific reporter probe with a fluorescent barcode via a capture probe. Digital counts from the barcodes are analysed using the nCounter analysis system and mapped back to the region of interest (ROI), providing spatial information about the targets within the ROI.

**Figure 3 cancers-12-02572-f003:**
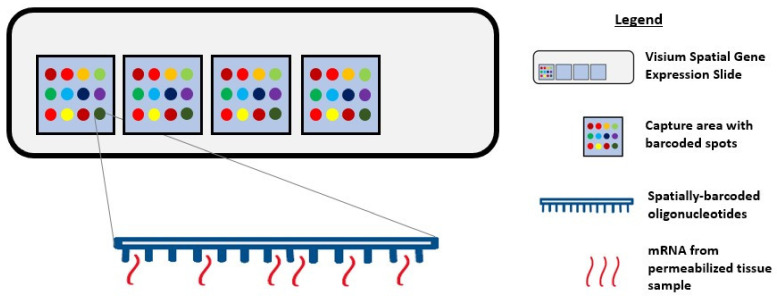
Diagram showing Visium Spatial Gene Expression Solution. Tissue samples are sectioned and placed in the 4 capture areas on the Visium Spatial Gene Expression slide. Each capture area contains over 5000 barcoded spots and each spot has multiple spatially barcoded oligonucleotides that will bind to mRNA released from permeabilized tissue samples.

**Figure 4 cancers-12-02572-f004:**
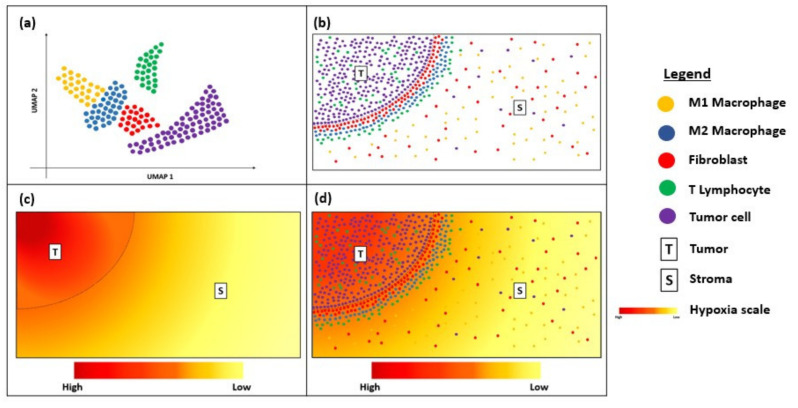
Diagram showing uniform manifold approximation and projection (UMAP) and hypoxia gradient in a tissue sample. (**a**) UMAP showing the immune landscape of tumour micro-environment; (**b**) Immune cells surrounding the tumour border; (**c**) Hypoxia gradient of tumour micro-environment; (**d**) Overlay of UMAP and the hypoxia gradient to visualize heterogeneity of the tumour micro-environment.

**Table 1 cancers-12-02572-t001:** Overview and comparison of the different imaging modalities.

Characteristics	Imaging Modalities			
	RNAscope	MERFISH	DSP	Visium
Vendor	Advanced Cell Diagnostics	Vizgen	NanoString	10× Genomics
Year of Launch	2012	2015–2016	2018–2019	2019
Target molecules	RNA	DNA or RNA	RNA or protein	mRNA
Methodology	Hybridization of branched DNA probes followed by signal amplification	Hybridization of fluorochrome-labelled barcoded DNA probes followed by signal amplification	Oligonucleotide-tagged RNA or antibody probes followed by photocleavage and sequencing or hybridization	On-slide cDNA barcoding followed by sequencing
FFPE validation	√	×	√	×
Maximum multiplexed capacity	12 RNA species	10,000+ mRNA species	96 proteins and 1400+ RNA	100,000+ unique molecule identifiers
Turnaround time	30 slides/11–14 h	1 slide/2–3 days	10–20 slides/48 h	4 capture areas/slide/day
Whole slide imaging	√	×	Possible but very costly and time consuming	√
Resolution	<1 μm	<1 μm	10 μm	55 μm
Key equipment required	Standard bright-field or fluorescence microscope	Microscope integrated with an automatic fluid handling system	Pressure cooker (for manual slide prep) or automated stainer (for Leica slide prep), GeoMx Digital Spatial Profiler	Cryostat, microscope, sequencer
Analytic software	HALO^®^	3D-daoSTORM or MERlin (for decoding)	Bundled software	Bundled software
Commercialized	√	In progress	√	√
Cost	$$	$	$$$	$$
Publications	2000+	10+	15+	4
Advantages	– End-to-end workflow available and can be automated – Easily assimilated into existing lab infrastructure – Single-cell spatial resolution – High sensitivity (able to detect low abundance RNA) – Whole tissue analysis – FFPE compatible	– Single-cell and subcellular spatial resolution – High multiplexing capacity (~10,000 mRNA) – High sensitivity (able to detect low-abundance RNA) – Whole tissue analysis	– End-to-end workflow available and automated – Easily assimilated into existing lab infrastructure – Able to profile both protein and RNA – Integrity of tissue sample is preserved, allowing reuse of sample – FFPE compatible	– End-to-end workflow available and automated – Easily assimilated into existing lab infrastructure – Full transcriptome analysis – Whole tissue analysis
Disadvantages	– Limited multiplexed capacity (~12 RNA species) – Only profiles RNA	– Complex workflow that requires extensive probe design and downstream validation – Requires installation of specialized equipment – Only profiles DNA or RNA – Tissues are fixed with paraformaldehyde, and no FFPE validation yet	– Poor single-cell spatial resolution – Limited multiplexed capacity (~1400 RNA and 96 proteins) – Requires the selection of a region of interest (ROI), causing biased hypothesis-driven analysis – Whole slide imaging only allows for the visualization of up to four markers	– Single-cell spatial resolution not yet achieved – Relatively lower sensitivity than RNAscope and MERFISH – Only profiles RNA – Requires fresh frozen tissue, and is not compatible with FFPE tissues – An optimization test is required for every new tissue

Abbreviations: DSP, digital spatial profiling; FFPE, formalin-fixed paraffin-embedded.

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
