# Peer review of "Transcriptional Spatial Profiling of Cancer Tissues in the Era of Immunotherapy: The Potential and Promise"

_cancers, 2020, doi:10.3390/cancers12092572_

Round 1
Reviewer 1 Report
This review offers a well-synthesized presentation of some of the most often used technologies for spatial transcriptomic profiling of solid tumors and how these technologies were leveraged to advance our understanding of the interplay between microtumor environment and disease progression or response to immunotherapy. Nerurkar and colleagues present a well-written piece and that will be of interest to both physicians and clinical researchers. My only concern lies with the section on digital spatial profiling, which would require clarification as per the comments below. Minor comments related to other sections of the text are also noted here.
Specific comments:
In the first introductory paragraph the authors state that. “The reason for this poor result in a subset of patients is largely attributed to the composition of the dynamic tumour microenvironment”. Although the tumour microenvironment has been linked to immunotherapy response, I am not aware of a particular study that would have established the tumour microenvironment as the main contributor (“largely attributed”) to immunotherapy failure. A citation for such work, if it exists, is required. Else, the statement should be softened.
Lines 84-86: the authors should specify the range of the number of probes that can be used at once with multiplex smFISH because “a large number of RNA” is only large relative to something else.
Lines 142-143 state that “we ultimately need FISH-based technologies that permit a high-throughput transcriptomic analysis” but do not give a reason for why such a technology is needed. The authors should explain why they think this is the next necessary step here.
Line 237: Is there a word missing? “to explore responses (of?) Non-Small-cell…”
Lines 256-260: The information pertaining to the poor single-cell resolution of DSP seems at odd with the following sentence where it is said that DSP requires profiling every cell at single-cell resolution. How can the technique require single-cell profiling if the technique provides poor single-cell resolution and a minimum of 10 cells in a ROI to give a good signal?
It is not clear from the text and Figure 2 how DSP delivers spatial information and how the signal is detected. It is also my understanding that with DSP target RNA is detected using RNA probes, not antibodies, which is not clear from the text and for which Figure 2 is somewhat misleading.
Author Response
Response to Reviewer 1 Comments
Point 1: In the first introductory paragraph the authors state that. “The reason for this poor result in a subset of patients is largely attributed to the composition of the dynamic tumour microenvironment”. Although the tumour microenvironment has been linked to immunotherapy response, I am not aware of a particular study that would have established the tumour microenvironment as the main contributor (“largely attributed”) to immunotherapy failure. A citation for such work, if it exists, is required. Else, the statement should be softened.
Response 1: We thank the reviewer for highlighting this. We have softened the statement as suggested by replacing “largely attributed” with “is partly attributed” in lines 40-41 as we were unable to find a specific study that highlights the tumour microenvironment as a main contributor to immunotherapy failure. We have also added in 3 references [8-10] to support our statement on the role of tumour microenvironment in immunotherapy response.
Point 2: Lines 84-86: the authors should specify the range of the number of probes that can be used at once with multiplex smFISH because “a large number of RNA” is only large relative to something else.
Response 2: We thank the reviewer for highlighting this. We have added the range of the number of RNA molecules that can be simultaneously detected per cell in lines 85-86 (“detect approximately 10, 000 genes and around 70, 000 – 100,000 RNA molecules per cell”). There are two numbers involved: i) the number of genes that can be studied in one experiment (~10,000 genes), ii) the number of individual RNA molecules that can be studied (~70,000-100,000 molecules), as cells express multiple RNA transcripts per gene. We have put both numbers to make it clearer to understand the difference.
Point 3: Lines 142-143 state that “we ultimately need FISH-based technologies that permit a high-throughput transcriptomic analysis” but do not give a reason for why such a technology is needed. The authors should explain why they think this is the next necessary step here.
Response 3: We thank the reviewer for the suggestion. We have added in the reason why high-throughput transcriptomic analysis is needed in lines 142-143 (“to better characterize rare populations of cells and cell types that display a distinctive gene expression profile”) A genome-wise transcriptomics allows researchers to gain additional insight about cellular and tissue dynamics and address exciting questions. In particular, by studying more cells and more genes at the same time, it is now possible to characterize rare populations of cells that may show distinctive gene expression, which is otherwise overlooked in non-multiplex platform.
The turnaround time of MERFISH was also changed from 48 hours to “2-3 days” in Line 186 Table 1 as the time varies depending on the experiment setup.
Point 4: Line 237: Is there a word missing? “to explore responses (of?) Non-Small-cell…”
Response 4: We thank the reviewer for highlighting this error. We have corrected it to “explore responses of Non-Small-cell Lung Cancer (NSCLC)” in line 263 accordingly.
Point 5: Lines 256-260: The information pertaining to the poor single-cell resolution of DSP seems at odd with the following sentence where it is said that DSP requires profiling every cell at single-cell resolution. How can the technique require single-cell profiling if the technique provides poor single-cell resolution and a minimum of 10 cells in a ROI to give a good signal?
Response 5: We thank the reviewer for highlighting this. We have removed the conflicting statement (lines 283-284) as DSP does not require profiling at single-cell resolution.
Point 6: It is not clear from the text and Figure 2 how DSP delivers spatial information and how the signal is detected. It is also my understanding that with DSP target RNA is detected using RNA probes, not antibodies, which is not clear from the text and for which Figure 2 is somewhat misleading.
Response 6: We thank the reviewer for highlighting this. We have elaborated further in lines 195-207 (“Oligonucleotide tags are conjugated using a photocleavable UV light-sensitive linker to primary antibodies or RNA probes, for protein or RNA profiling, respectively. Firstly, the FFPE tissue section undergoes antigen retrieval and incubation with oligo-labelled primary antibodies or proteinase K digestion and incubation with an RNA probe cocktail. The slides are then stained with fluorescently labelled antibodies (up to 4 markers) to visualize features of interest and enable region of interest (ROI) selection on the tissue sample. The user-defined ROI is illuminated with UV light, releasing the photocleavable oligonucleotides from the antibodies or RNA probes. These oligonucleotides are collected in a microcapillary tube and transferred to a microtiter plate for quantitation. This process is repeated for the next ROI. Once all ROIs have been processed, spatially resolved pools of oligonucleotides are hybridized to capture and reporter probes with unique fluorescent barcodes that can be digitally counted using the nCounter analysis system. Alternatively, Next-generation sequencing (NGS) may be used as a readout where reads are processed into digital counts that are mapped back to the ROI to allow spatial profiling of the ROI.”) We have elaborated on how DSP delivers spatial information and signal detection by explaining how all samples are stained with antibody-based markers to visualize tissue morphology and enable region of interest (ROI) selection, followed by collection of photocleavable oligonucleotides from the defined ROI. In terms of signal detection, the spatially resolved oligonucleotides can be hybridized to reporter probes with barcodes and analyzed using the nCounter analysis system. Alternatively, they can be quantified by next-generation sequencing (NGS) and mapped back to the ROI to allow visualization of targets within the chosen ROI.
We have also revised Figure 2 and its description in lines 218-228 (“Diagram showing Digital Spatial Profiling (DSP) for detection of RNA targets (a) After the FFPE tissue sample undergoes proteinase K digestion, it is incubated with RNA probes that bind to target RNA in the sample. Each RNA probe is conjugated to an oligonucleotide tag via an ultraviolet (UV) photocleavable linker; (b) The slide is stained with 3 antibody-based markers (up to a maximum of 4 markers) and imaged to visualize tissue morphology and enable selection of regions of interest (ROI); (c) Illumination of the ROI with UV light causes the release of photocleavable oligonucleotides from the RNA probes. The oligonucleotides are collected in a microcapillary tube and transferred to a microtiter plate; (d) Spatially resolved pools of oligonucleotides bind to a target-specific reporter probe with a fluorescent barcode via a capture probe. Digital counts from the barcodes are analyzed using the nCounter analysis system and mapped back to the ROI, providing spatial information about the targets within the ROI.”) to focus on DSP for detection of RNA targets and replaced the antibodies with RNA probes as suggested.
We would like to thank the reviewer for the comments. We have also attached a PDF file with the above point-to-point responses.

Reviewer 2 Report
The submitted manuscript with the title :"Transcriptional spatial profiling of cancer tissues in 2 the era of immunotherapy: The potential and promise" is one of the few tries to summarize and catalogize the currently available techniques in the newly emerged field of spatial profiling. The manuscript is well written and comphrehensively structured which leaves little room for criticism.
Only remarks from my side are:
- Page 2, line 62 onwards: Please take note and comment on the layers of biology. Not every transcript leads to translation and not every protein is necessarily biologically active with respect to phosphorylation/kinase activity. Just because an mRNA or a protein is expressed it does not have to exert biological impact.
- Spatial transcriptomics originated from the RNA in situ sequencing efforts at the SciLifeLab / KTH/ KI in Stockholm/Uppsala in the groups of Stahl/Frisen/Lundeberg/Nilsson. As the spatial transcriptomics is a main part, please refer to the major early pulications from these investigators.
- Genome-wide spatial profiling on FFPE tissues has been recently described on BiorXiv
Author Response
Response to Reviewer 2 Comments
Point 1: Page 2, line 62 onwards: Please take note and comment on the layers of biology. Not every transcript leads to translation and not every protein is necessarily biologically active with respect to phosphorylation/kinase activity. Just because an mRNA or a protein is expressed it does not have to exert biological impact.
Response 1: We thank the reviewer for highlighting this point. We have added in the suggested points in lines 419-427 (“Thus, while transcriptional spatial profiling techniques may enable extensive sequencing up to the entire genome level, one must consider that not every mRNA transcript necessarily leads to translation and synthesis of biologically active proteins that contribute to tumor initiation, progression and therapeutic response. This may be attributed to the heterogeneity in signaling pathways, post-translational modifications and protein isoforms which limits the utility of mRNA abundance as a proxy for protein abundance and activity. Therefore, protein-based technologies such as multiplex IHC/IF remain integral for downstream analysis after transcriptomics studies as they serve as an excellent endpoint to validate protein function after identifying genetic targets.”) to highlight the different layers of biology clearly.
Point 2: Spatial transcriptomics originated from the RNA in situ sequencing efforts at the SciLifeLab / KTH/ KI in Stockholm/Uppsala in the groups of Stahl/Frisen/Lundeberg/Nilsson. As the spatial transcriptomics is a main part, please refer to the major early pulications from these investigators.
Response 2: We thank the reviewer for highlighting these publications. We have added in the relevant publications from these authors into the text – references (88), (101), (106) and (107).
Point 3: Genome-wide spatial profiling on FFPE tissues has been recently described on BiorXiv https://doi.org/10.1101/2020.07.24.219758
Response 3: We thank the reviewer for highlighting this paper. We have added this paper to the text in lines 386-388 (“Furthermore, although Visium has only been validated in fresh frozen specimens, early studies suggest that it can be used for genome-wide spatial profiling in FFPE specimens as well.”) to highlight that although Visium has only been validated in fresh frozen tissues, there are early studies suggesting its use can be extended to FFPE tissues.
We thank the reviewer for the comments. We have also attached a PDF file of the above point-to-point responses.
